# Does quality of antenatal care influence antepartum stillbirth in Hossana City, South Ethiopia?

**TrhasTadesse Berhe**[1]*, **Lebitsi Maud Modibia**[2], **Addisu Tadesse Sahile**[3], **Getachew Woldeyohanes Tedla**[1]

**1** Yekatit 12 Hospitals, Department of Public Health, College of Medicine, Addis Ababa, Ethiopia, **2** The University of South Africa, Department of Health Studies College of Human Sciences, Pretoria, South Africa, **3** Department of Public, Health Unity University, Addis Ababa, Ethiopia

* ttrhas@gmail.com

**Data Availability Statement:** The finding of this study is generated from the data collected and analyzed based on the stated methods and materials. There are supplementary files. The

## Abstract

### Background

Antepartum stillbirth is a public health problem in a low-income country like Ethiopia. Quality antenatal care (ANC) is supposed to reduce the risk of many bad outcomes. Thus the main objective of this study was to identify the effect of quality antenatal care on antepartum stillbirth in Public health facilities of Hossana town Hadiya zone south Ethiopia.

### Method

About 1123 mothers with a gestational age of less than 16 weeks were identified and followed using an observational longitudinal study to determine whether the quality of ANC influences antepartum stillbirth or not. Standardized and pretested observation checklists and participants' interview questionnaires were employed to obtain the necessary information after getting both written and verbal consent from the concerned bodies and study participants. In this study, quality was measured by the process attributes of quality to measure the acceptable standard of quality of antenatal care. Women who received ≥75% of essential ANC services (from 1st-4th visit) were categorized under received good quality antenatal care. General estimating equation analysis was done to determine the effect of quality antenatal care on antepartum stillbirth.

### Result

A total of 121 (12.3%) 95% CI (10.3%, 14.5%) mothers who were observed during delivery had encountered antepartum stillbirth. In this study, the overall quality of antenatal care service that was provided in the whole visit (1st -4th) was 1230 (31.38%). Higher quality ANC decreases the odds of antepartum stillbirth by almost 81%, after controlling other factors (0.19 (AOR 0.19 at 95% CI; 0.088 to 0.435). There is a change in the odds of developing antepartum stillbirth as the level of education of mothers increases. Moreover, mothers with a history of preexisting hypertension were more like to have antepartum stillbirth AOR = 3.1, 95%CI (1.44, 6.77)].

original data supporting this finding will be available at any time upon request.

**Funding:** TTrhas Tadesse Berhe received the grant from Yekatit 12 Hospital Medical College, Addis Ababa, Ethiopia. The funder (Yekatit 12 Hospital Medical College) covers only the transport and per diem cost of data collectors and supervisors during the data collection process. The funders had no role in the study design, analysis, decision to publish, or preparation of the manuscript except covering the aforementioned costs during the data collection period.

**Competing interests:** The authors have declared that no competing interests exist.

## Conclusion and recommendation

Therefore, having a good quality of ANC significantly reduces antepartum stillbirth. Strategies need to be developed on the problems identified to improve the quality of ANC and reduce antepartum stillbirth significantly.

## Introduction

Pregnancy and childbirth are times of happiness for parents and families [1, 2]. However, 2.6 million pregnancies end with stillbirths annually [3]. Among stillbirths, 98% of them occur in developing countries. Fifty percent of all stillbirths occur during antenatal care [4] Most of these deaths result from preventable conditions [5].

Sub-Saharan Africa (SSA) has the highest stillbirth rate globally with 28.3 per 1000 births, compared to 3.1 for high-income countries. This accounts for about a third (35.4%) of the global burden of stillbirths [6].

Like in many countries in SSA, stillbirths are not routinely and adequately recorded and monitored in Ethiopia [7]. The estimates of the stillbirth rate from the study conducted in Gonder University Hospital, North West Ethiopia was 71 per 1,000 total births and this study revealed that lack of ANC was the main contributor to stillbirth [8].

HIV infection, malaria, malnutrition, and anemia during pregnancy along with maternal age are some factors causing stillbirth. In addition, being a primigravida or having stillbirth or miscarriage in earlier pregnancies, unplanned pregnancy, and short inter-pregnancy spacing, chronic diseases, obesity, low socio-economic status, alcohol, and other drug use have been reported to be risk factors for, stillbirth and neonatal mortality [8–10]. It was indicated in most studies that quality of care is likely to be the key factor for stillbirth [11–13]. Quality ANC is considered to improve maternal and infant health significantly and it also reduces maternal and prenatal morbidity [1]. However, the effectiveness of specific ANC quality interventions as a means of reducing adverse birth outcomes like antepartum stillbirth in Ethiopia has not been rigorously evaluated [14]. Therefore, it was found important to conduct this study on the effect of quality antenatal care on the birth outcome (antepartum stillbirth). Where such a study may be a help to guide in designing an action plan on how antepartum stillbirth is to be reduced in developing countries like Ethiopia.

## Method and material

### Study area, design, and population

Health facility-based observational longitudinal was conducted from July 2017 to July 2018 in Hossana City, South Ethiopia. From previous research outcomes published in Hindawi Journal [15], among 1123 mothers who participated in the study, about 980 completed the follow-up and know their status whether they have live birth or not.

The study areas for this research were all of the government health facilities which provide ANC & delivery services in Hossana Town (one zonal referral teaching hospital (Nigist Eleni Teaching Hospital) and three health centers, namely, Hossana, Bobicho, and Lich Amba Health Centre), Hadiya Zone, and South Ethiopia.

Mothers with a gestational age of less than 16 weeks were identified and followed using an observational longitudinal study to determine whether the quality of ANC influences antepartum stillbirth or not. Pregnant mothers visiting for ANC services in Hosanna town during the

study period and those who full fill the inclusion criteria (mothers who accepted consent and mothers who were in the age range of 18 and above) were taken as the study population.

## Sample size and sampling technique

The current study is part of a large follow-up study with multiple objectives. The detail of the sample size calculation assumptions to address all objectives is described in the other part of the study published on Hindawi, advanced public health [15]. For the current study, the sample size was calculated with the prevalence of antepartum stillbirth (7.1%) among women with ≥4ANC visits [16] considering these mothers as they received good quality ANC service. 80% power, 95% confidence level, a ratio of unexposed (women who received poor quality ANC service) to exposed (women who received good quality ANC service) is 2:1and 20% non-response rate. The final calculated sample size was 936. However, to increase the power of the study, all 980 women who attended ANC services were included in the analysis. Those women who died during pregnancy and delivery, and had an abortion were excluded from the study.

## Data collection tool and procedure

Four B.Sc. midwives and two female MSc midwifery nurses who were not staff members in the health facilities were recruited as data collectors and supervisors respectively. Data were collected through a standard structured observation checklist adopted from a maternal and child health integrated program to assess the quality of ANC [15]. The quality of ANC service in each four focused ANC visits was measured by history taking, physical examination, diagnosis and management, and counseling In addition, there was also a checklist about the status of the baby at birth while they were visiting the health facilities for delivery of their baby. If women did not visit the health facilities for delivery of their baby, the data collectors were tracing them at home, based on their address registered during the first visit.

## Variables and their operational definition

**Dependent variable.** *Antepartum stillbirth*. For mothers who gave birth at a health facility, operationalized as delivery of any non-viable fetus after 27 weeks of gestation, or with a birth weight more than 500 g, with an Apgar score of 0 at 1 and 5 min and signs of maceration, or absent fetal heart sound before the initiation of labor [17] and if the mother gave birth at home and delivered with stillbirth it was considered to be antepartum stillbirth [18]. Hence, had the mothers obtained good quality antenatal care, they would have delivered in health facilities since birth preparedness and complication readiness is one of the components of focused ANC services.

*Independent*. The quality of ANC service was the primary exposure of interest. The other exposure variables include a place of delivery, parity, residence, educational status, maternal occupation, marital status, maternal age, income, History of premature delivery, history of stillbirth, Pre-existing HTN, and DM. In this study, quality was measured by the process attributes of quality to measure the acceptable standard of quality of antenatal care. Women who received ≥75% of essential ANC services (from 1st-4th visit) are categorized under received good quality antenatal care.

*Data analysis*. The data entry and analysis were made using Epi-info and STATA version 14 respectively. Socio-demographic variables and other factors like the medical condition of the mother were analyzed using descriptive statistics. Generalized estimating equation analysis with binary response variable using robust estimator and exchangeable working correlation matrix was carried out to control the cluster effect of the data among women who received ANC services within the same facility by the same ANC provider. Based on Hosmer and

Lemeshow applied logistic regression guide a p-value < 0.2 was considered to select eligible variables for the final model and a p-value < 0.05 was considered to identify statistically significant predictor variables for antepartum stillbirth.

## Quality control

The training was given to data collectors and supervisors on the aim of the research, the content of the questionnaire, and how to conduct it. The collected data were checked every day by supervisors and the principal investigator for their completeness and consistency.

## Ethical considerations

This study was a component of Strategies for quality antenatal care to improve birth outcome, which received institutional review board approval from the University of South Africa (UNISA reference number (UNISA-ET/KA/ST/29/14-06-17).), and from Southern nations and nationalities people's region Ethiopian Health Bureaus and Hadiya Zone. Both ANC clients and providers were informed about the purpose of the study and verbal and written informed consent was also obtained before data collection by explaining the goals of the study, the credibility of the researchers' confidentiality, anonymity, and the freedom to opt-out of the study at any stage without negative consequences. Furthermore, report writing didn't refer to a specific respondent with identifiers

## Result

### Background characteristics of the study participants

About 980 (87.3%) completed the follow-up (from their first ANC visit to the delivery of the baby). A significant number of 765 (78.1%) of the women who participated in the study were found between 20–34 years of age and greater than 34 were found to be 34(3.5%). The mean age of respondents was 25.6 years (SD ± 4.1), with a median of 25 and with minimum and, maximum ages of 18 and 39 respectively (Table 1).

### Past obstetric history of the respondents

The mean (standard deviation) pregnancy was 3.5 (1.9) ranging from 1 to 10. About 412 (42.0%) of the mothers experienced pregnancy two to four times. While nearly 349(35.6%) have been pregnant more than four times in their lifetime. More than two third of the total mothers 725(74%) have experienced birth at least once. Whereas nearly 485 (66.9%) of the mothers gave birth two to four times before the current study and only 35(4.8%) of them gave birth greater than four times.

Of the total 725(74%) mothers, who have previous delivery, only 73(10.1%) of the study participant knew the birth weight of their last baby, 155(15.8%) had a history of premature delivery, 277(28.3%) had a history of stillbirth and 23(24.8%) had a history of neonatal death at least one times in their lifetime. Almost half of 445 (45.4%) experienced abortion at least once (Fig 1).

### Medical history of the respondents

Regarding the medical history of the pregnant women, only 9(0.9%) and 10(1%) had a history of mental health problems and cardiac disease problems respectively. In the current finding, 145(14.8%) of the study participants had BMI greater than 30kg/ m2, and the rest majority 835 (85.2%) were in the range between 16 and 30 body mass index at the beginning of the pregnancy. About (8.1%) of the mothers had a history of diabetic Mellitus while 66(6.7%) had a history of hypertension (Fig 2).

**Table 1. Socio-demographic characteristics of the participant (n = 980) at Hossana town public health facilities, Southern Ethiopia July 2017 June 2018.**

| | Frequency | Percentage |
|---|---|---|
| Age (in year) | | |
| <20 | 181 | 18.5 |
| 20–34 | 765 | 78.0 |
| >34 | 34 | 3.5 |
| **Marital Status** | | |
| Married | 947 | 96.4 |
| Never married and divorced | 33 | 3.6 |
| Religion | | |
| Protestant | 552 | 56.3 |
| Orthodox | 187 | 19.1 |
| Muslim | 118 | 12 |
| Hawariyat | 117 | 11.9 |
| Catholic | 6 | 0.6 |
| Ethnicity | | |
| Hadiya | 570 | 58.2 |
| Kenbata | 173 | 17.7 |
| Amhara | 80 | 8.2 |
| Silitie | 76 | 7.8 |
| Others (Tigrwiti, Guragi, Oroma & Cidama) | 81 | 8.3 |
| Educational status of the mother | | |
| Don't read and write | 85 | 8.7 |
| Read and write | 50 | 5.1 |
| Grade 1–6 | 164 | 16.7 |
| Grade 7–8 | 155 | 15.8 |
| Grade 9–10 | 242 | 24.7 |
| Grade 11–12 | 117 | 11.9 |
| Higher level12+ | 167 | 17.1 |
| Monthly income | | |
| <2000 ETB | 607 | 61.9 |
| ≥2000 ETB | 373 | 38.1 |
| Occupation | | |
| House wife | 464 | 47.3 |
| Daily labourer | 148 | 15.1 |
| Government employee | 143 | 14.6 |
| Merchant | 134 | 13.7 |
| Others (farmer, pastor & private) | 91 | 9.3 |
| Place of residences | | |
| Urban | 667 | 69 |
| Rural | 304 | 31 |

## The magnitude of quality of ANC service and stillbirth

The overall magnitude of quality of ANC in the whole visit was 1230 (31.38%) at 95% CI [28.8, 34.7]. Concerning birth status, a total of 121 (12.3%) 95% CI (10.3%, 14.5%) of the mothers delivered stillbirth, and 838 (87.7%) were alive babies (Fig 3).

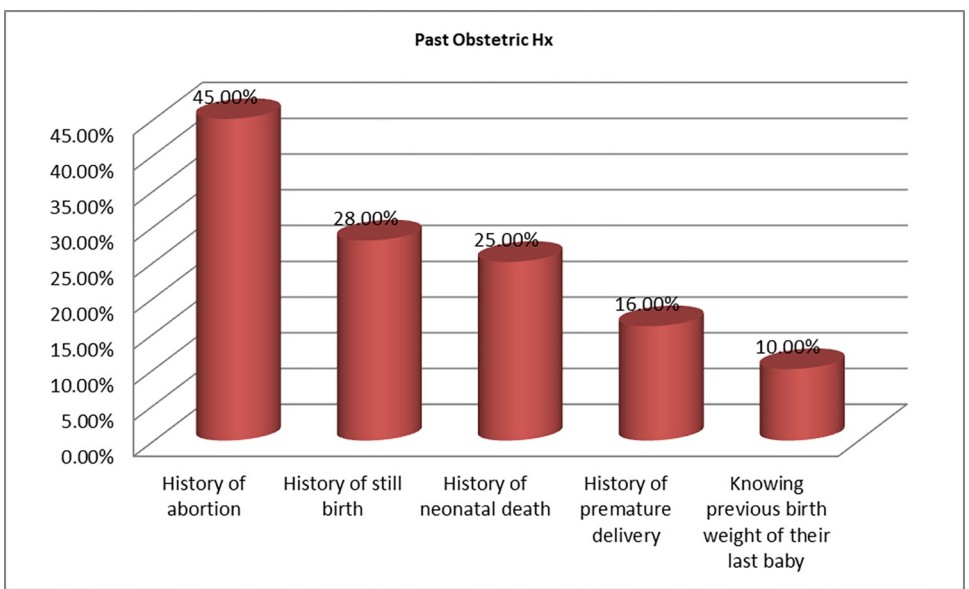

**Fig 1. Previous obstetric history of the study participant (n = 980) at Hossana town public health facilities, SNNPR, Ethiopia July 2017 June 2018.**

## Association between stillbirth and quality of ANC

After initial exploratory analysis, the independent and multiple variable effects of explanatory variables were assessed. Variables entered in the multivariable analysis that had a significant association (p-value<0.2) in the binary general estimating equation and those of known clinical relevance and from previous publications. These included maternal age, parity, occupation of the mother, education status of the mother, place of birth, body mass index at the first visit, history of mental health problems, pre-existing hypertension, pre-existing diabetes, cardiac disease, previous stillbirths, previous premature delivery, previous abortion, and quality of ANC.

In a Generalized Estimating Equation binary logistic regression analysis, antepartum stillbirth had a positive association with poor antenatal care quality, being rural residence, being

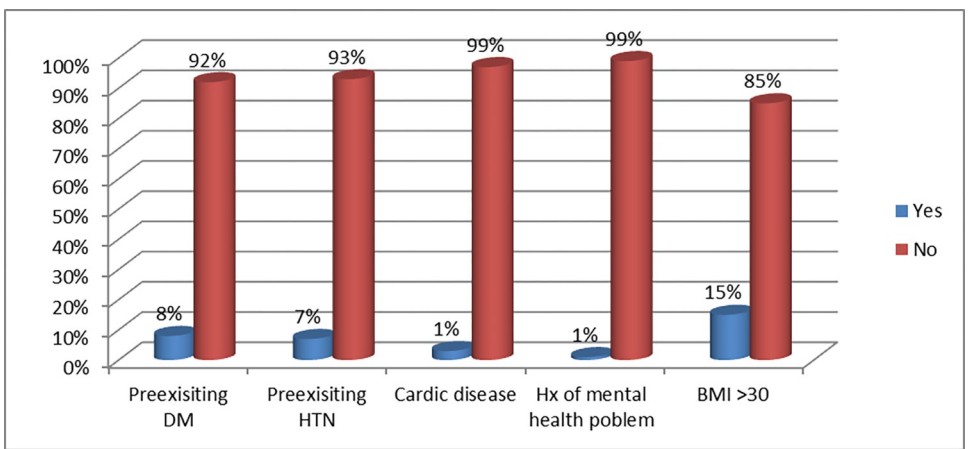

**Fig 2. Previous medical history and BMI at 1st visit of the study participant (n = 980) at Hossana town public health facilities, SNNPR, Ethiopia July 2017 June 2018.**

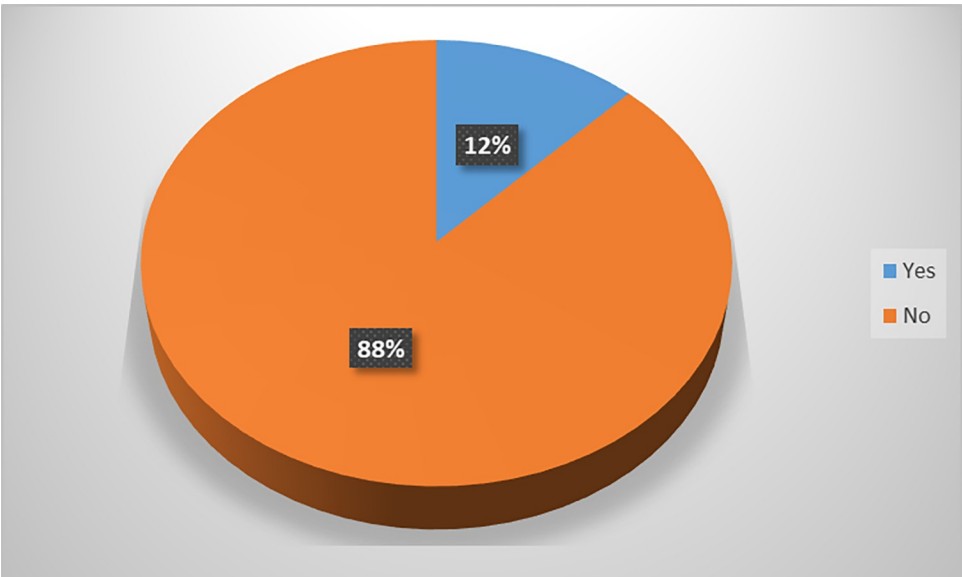

**Fig 3. Magnitude of stillbirth of the study participant (n = 980) at Hossana town public health facilities, SNNPR, Ethiopia July 2017 June 2018.**

age less than 20, being merchant a government employee, being never married and divorced, being illiterate, being greater than 4 delivery, history of premature delivery, history of stillbirth and history of hypertension. The other socio-demographic backgrounds of the study women (, ethnicity, religion, and income) were not associated with stillbirth. In multivariable Generalized Estimating Equation logistic regression analysis antenatal care quality, educational status, and history of hypertension remained to have a statistically significant association with antepartum stillbirth. Higher quality ANC decreases the odds of antepartum stillbirth by almost 81%, after controlling other factors (0.19 (AOR 0.19 at 95% CI; 0.088 to 0.435. Moreover mother with educational status read and write, grade 1–6, grade 7–8, grade 9–10 and grade 11–12 were more likely to develop antepartum stillbirth compared to mothers with educational status with a higher level [AOR = 15.9, 95%CI (1.89,34.72)],[AOR = 5.2, 955CI (2.04,13.18)], AOR = 4.1, 95%CI (1.72,9.89)], AOR = 5.9, 95%CI (2.45,14.58)] and AOR = 3.4, 95%CI (1.38,8.25)] respectively. Similarly, mothers with a history of preexisting hypertension were more like to have antepartum stillbirth AOR = 3.1, 95%CI (1.44, 6.77)] (Table 2).

## Discussion

In the current study, higher quality of ANC decreases the odds of antepartum stillbirth by almost 81%, after controlling other factors (0.19 (AOR 0.19 at 95% CI; 0.088 to 0.435). This finding is consistent with the study conducted in Kassala, Eastern Sudan, which acknowledged that the use of quality ANC especially administering iron sulfate supplements for three months and the use of a bed net for prevention of malaria during pregnancy were the most protective factors for perinatal mortality (stillbirth and early neonatal death) [19].

This study also goes in line with the study conducted in Gonder Hospital, Ethiopia, in which it was reported that women who had not had quality ANC follow-up were more likely to have antepartum stillbirth compared to women who had a good quality of ANC [8]. The finding of this study is also consistent with the study conducted in Ethiopia using the

**Table 2. Generalised estimating equation logistic regression to identify determinants of antepartum stillbirth among pregnant women attending ANC at public health facilities of Hossana Town (n = 980), July 2017 to June 2018.**

| Variables | Antepartum Still birth | | COR(95%CI) | AOR(95%CI) |
|---|---|---|---|---|
| | Yes | No | | |
| **Quality ANC** | | | | |
| Poor quality | 113 | 560 | 0.14(0.728,0.275)* | 0.19(0.088,0.435)** |
| Good quality | 9 | 299 | 1 | |
| **Residence** | | | | |
| Urban | 77 | 559 | 1.3(0.884,1.960)* | |
| Rural | 44 | 260 | 1 | |
| **Age** | | | | |
| <20 | 29 | 152 | 0.33(0.074,1.444)* | 0.22(0.039,1.251) |
| 20–34 | 90 | 675 | 0.47(0.110,1.991) | 0.46(0.086,2.471) |
| >34 | 2 | 32 | 1 | 1 |
| **Occupation** | | | | |
| House wife | 64 | 400 | 1.2(0.668,2.279) | 1.5(0.684,3.136) |
| Merchant | 7 | 127 | 3.6(1.397,9.181)* | 2.7(0.892,8.271) |
| Daily labourer | 26 | 122 | 0.9(0.461,1.860) | 0.64(0.256,1.597) |
| Gov. employee | 9 | 134 | 2.9(1.227,7.038)* | 5.6(1.845,17.302) |
| Others* | 15 | 76 | 1 | 1 |
| **Marital status** | | | | |
| Never married & divorced | 1 | 32 | 4.6(0.628,34.329)* | 4.1(0.459,3.136) |
| Married | 120 | 827 | 1 | 1 |
| **Educational status** | | | | |
| Illiterate | 17 | 68 | 0.6(0.302,1.217)* | 2.3(0.926,5.535) |
| Read and write | 1 | 49 | 7.4(0.975,56.666)* | 15.9(1.89,34.72)** |
| Grade 1–6 | 14 | 150 | 1.6(0.800,3.300)* | 5.2(2.042,13.180)** |
| Grade 7–8 | 27 | 128 | 0.7(0.390,1.326) | 4.1(1.72,9.89)** |
| Grade 9–10 | 19 | 223 | 1.8(0.931,3.407)* | 5.9(2.45,14.58)** |
| Grade 11–12 | 21 | 96 | 0.7(0.362,1.331) | 3.4(1.376,8.254)** |
| Higher level(12+) | 22 | 145 | 1 | 1 |
| **Income** | | | | |
| <2000 (Ethiopian birr) | 75 | 532 | 0.99(0.674,1.477) | |
| ≥2000ETB | 46 | 327 | 1 | |
| **Gravidity** | | | | |
| 1 | 16 | 203 | 2.2(1.233,4.001) | |
| 2–4 | 53 | 359 | 1.2(0.785,1.791) | |
| >4 | 52 | 297 | 1 | |
| **Parity** | | | | |
| 1 | 23 | 182 | 0.23(0.030,1.784)* | 0.4(0.043,2.993) |
| 2–4 | 67 | 418 | 0.18(0.024,1.137)* | 0.2(0.026,1.659) |
| >4 | 1 | 34 | 1 | 1 |
| **History of abortion** | | | | |
| No | 60 | 400 | 0.9(0.605,1.297) | |
| Yes | 61 | 459 | 1 | |
| **History of premature delivery** | | | | |
| No | 96 | 729 | 1.5(0.905,2.356)* | 1.4(0.703,2.611) |
| Yes | 25 | 130 | 1 | 1 |
| **History of stillbirth** | | | | |

*(Continued)*

**Table 2.** (Continued)

| Variables | Antepartum Still birth | | COR(95%CI) | AOR(95%CI) |
|---|---|---|---|---|
| | **Yes** | **No** | | |
| No | 80 | 625 | 1.5(0.985,2.200)* | 0.9(0.516,1.640) |
| Yes | 43 | 234 | 1 | 1 |
| **History of neonatal death** | | | | |
| No | 90 | 647 | 1.1(0.679,1.627) | |
| Yes | 31 | 212 | | |
| **Pre-existing hypertension** | | | | |
| No | 108 | 806 | 1.8(0.966,3.469)* | 3.1(1.44,6.77)** |
| Yes | 13 | 53 | 1 | 1 |

N.B. 1 = Reference Category

** = Significant at p-value <0.05 in multivariate analysis

* Significant at p-value <0.2 in bivariate analysis

demographic health survey data, which concluded that experiencing stillbirth had significantly associated with the utilization of quality ANC [20]. The finding is also similar to the study finding in Ethiopia, which indicated that promoting four or more ANC utilization decreases the risk of having a stillbirth [21]. Furthermore, a systematic review analysis was conducted in developing countries to develop strategies. In the same vein, a study conducted in a developing country recommended providing quality ANC for a pregnant woman specifically reduction of infections through early detection and treatment of cases (like syphilis, Malaria) and improvement of maternal nutritional status through counseling and health education during ANC visit were the best strategy to reduce the stillbirth rate in a developing country [22]. This is also supported by the study conducted in Namibia, which showed that lack of quality of ANC (especially on the supplementation of folic acid), maternal characteristics, medical condition, and obstetric complications were the modifiable risk factor for antepartum stillbirth. The authors concluded that providing quality ANC to a pregnant woman (especially assessment of all the maternal risk factors at the first ANC visit) and counseling maternal sleeping position during pregnancy [23] was the most important strategy for the reduction of the incidence of stillbirth [24]. Similarly, the study finding resonates with the study conducted at Jimma University Specialised Hospital, which indicated that having a quality of ANC during pregnancy prevents the likelihood of having an antepartum stillbirth. This study has also strongly recommended the providing of quality ANC, focused on the counseling on birth preparedness and complication readiness to increase the utilization of health facility delivery [25]. Besides, a cohort study conducted in a multi-ethnic English maternity population identified that fetal growth restriction also had the largest population attributable risk for stillbirth and was five-fold greater if it was not detected during antenatal than when it was (32.0% *v* 6.2%). In this study providing quality ANC is the cornerstone for reduction of the incidence of stillbirth [26] in the same context, this study was also in line with the cohort study conducted in New Zealand. The study indicated that accessing <50% of recommended antenatal visits was associated with a more than twofold increase in late stillbirth compared with accessing the recommended number of visits and those infants that were not identified as being SGA before birth/ interview were nine times more likely to be stillbirth compared with those that were. The authors concluded that providing quality ANC especially in detecting the gestational age is the key strategy for decreasing the incidence of antepartum stillbirth [27].

## Conclusion

In this study after controlling the confounding factors in the GEE, multivariate analysis showed that higher quality of ANC decreases the odds of antepartum stillbirth by almost 81%, after controlling other factors.

## Limitation of the study

Due to the inclusion of mothers who gave birth outside the health facility, it may have overestimated the magnitude of antepartum stillbirth.

## Recommendation

The health care provider should organize formal and informal meetings to disseminate educational information on stillbirth. And the strategies should be developed focusing on the problems identified to enhance the quality of ANC and reduce antepartum stillbirths.

## Author Contributions

**Conceptualization:** TrhasTadesse Berhe, Lebitsi Maud Modibia.

**Data curation:** TrhasTadesse Berhe, Lebitsi Maud Modibia, Addisu Tadesse Sahile, Getachew Woldeyohanes Tedla.

**Formal analysis:** TrhasTadesse Berhe, Lebitsi Maud Modibia, Addisu Tadesse Sahile, Getachew Woldeyohanes Tedla.

**Funding acquisition:** TrhasTadesse Berhe.

**Investigation:** TrhasTadesse Berhe, Lebitsi Maud Modibia.

**Methodology:** TrhasTadesse Berhe, Lebitsi Maud Modibia.

**Project administration:** TrhasTadesse Berhe.

**Resources:** TrhasTadesse Berhe.

**Software:** TrhasTadesse Berhe, Addisu Tadesse Sahile, Getachew Woldeyohanes Tedla.

**Supervision:** TrhasTadesse Berhe, Lebitsi Maud Modibia.

**Validation:** TrhasTadesse Berhe, Lebitsi Maud Modibia, Addisu Tadesse Sahile, Getachew Woldeyohanes Tedla.

**Visualization:** TrhasTadesse Berhe, Lebitsi Maud Modibia, Addisu Tadesse Sahile, Getachew Woldeyohanes Tedla.

**Writing – original draft:** TrhasTadesse Berhe, Lebitsi Maud Modibia, Addisu Tadesse Sahile, Getachew Woldeyohanes Tedla.

**Writing – review & editing:** TrhasTadesse Berhe, Lebitsi Maud Modibia, Addisu Tadesse Sahile, Getachew Woldeyohanes Tedla.

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
