## [Decision Letter · Decision Letter 0]

9 Aug 2022

PGPH-D-22-00340

DOES QUALITY OF ANTENATAL CARE INFULENCE ANTEPARTUM STILL BIRTH?IN HOSSANA CITY, SOUTH ETHIOPIA,

Dear Dr. Tedla,

Thank you for submitting your manuscript to PLOS Global Public Health. After careful consideration, we feel that it has merit but does not fully meet PLOS Global Public Health’s publication criteria as it currently stands. Therefore, we invite you to submit a revised version of the manuscript that addresses the points raised during the review process.

Please note that we have only been able to secure a single reviewer to assess your manuscript. We are issuing a decision on your manuscript at this point to prevent further delays in the evaluation of your manuscript. Please be aware that the editor who handles your revised manuscript might find it necessary to invite additional reviewers to assess this work once the revised manuscript is submitted. However, we will aim to proceed on the basis of this single review if possible. 

We look forward to receiving your revised manuscript.

Kind regards,

Julia Robinson

Executive Editor

Journal Requirements:

1. Please amend your detailed online Financial Disclosure statement. This is published with the article. It must therefore be completed in full sentences and contain the exact wording you wish to be published.

Please state the initials, alongside each funding source, of each author to receive each grant.

2. Please ensure that the funders and grant numbers match between the Financial Disclosure field and the Funding Information tab in your submission form. Note that the funders must be provided in the same order in both places as well.

3. Please update your online Competing Interests statement. If you have no competing interests to declare, please state: “The authors have declared that no competing interests exist.”

4. Please provide separate figure files in .tif or .eps format and remove any figures embedded in your manuscript file. Please also ensure that all files are under our size limit of 10MB.

5. We suggest you thoroughly copyedit your manuscript for language usage, spelling, and grammar. If you do not know anyone who can help you do this, you may wish to consider employing a professional scientific editing service. 

Additional Editor Comments (if provided):

Reviewers' comments:

Reviewer's Responses to Questions

**Comments to the Author**

1. Does this manuscript meet PLOS Global Public Health’s publication criteria? Is the manuscript technically sound, and do the data support the conclusions? The manuscript must describe methodologically and ethically rigorous research with conclusions that are appropriately drawn based on the data presented.

Reviewer #1: No

2. Has the statistical analysis been performed appropriately and rigorously?

Reviewer #1: No

3. Have the authors made all data underlying the findings in their manuscript fully available (please refer to the Data Availability Statement at the start of the manuscript PDF file)?

Reviewer #1: Yes

4. Is the manuscript presented in an intelligible fashion and written in standard English?

Reviewer #1: No

5. Review Comments to the Author

Reviewer #1: Manuscript title: Does quality of antenatal care influence antepartum still Birth? in Hossana city, south Ethiopia

The manuscript employs a longitudinal study design to investigate the influence of quality prenatal treatment on antepartum stillbirth in Hossana town, Hadiya zone, south Ethiopia. There were 1123 mothers studied. The study discovered that the quality of prenatal care was linked to antepartum stillbirth, with women who got poor quality ANC having a higher risk of stillbirth. The authors went on to argue that strategies for addressing the mentioned issues are needed to dramatically enhance the quality of ANC and prevent antepartum stillbirth. This work has validity in my opinion and would be of interest to readers of Plos Global Public Health. However, additional modifications would be required to make it scientifically suitable for publication consideration. My general and specific remarks are listed below.

General comments

In most areas of the work, I noticed some concerns with poor grammar and sentence organisation. The manuscript would need to be thoroughly edited. I've also noticed that the authors did not follow the journal's formatting guidelines for headings, line numbers, and reference style. This has made it more difficult for me to constantly offer particular places where problems have been detected.

Specific comments

Title

I think the title has some few issues. For example “influence” is wrongly spelled. I also think that the ‘?’ sign has been placed at the wrong place. If you need to include it in the title at all, then it could come at the end.

Abstract

The journal requires that the abstract is structured. I can see the authors have attempted to do that but the “Background” heading is missing.

“The odds of poor quality of ANC services among pregnant mothers 0.19 (AOR 0.19) at 95% CI; 0.088 to 0.435). Thus, those mothers who received poor quality of ANC services are 81% more likely to develop antepartum still birth compared to those mothers who received good quality of ANC services”.

I believe a revision is required. The first statement appears to have no meaning. The odds ratio's interpretation is similarly incorrect. An odds ratio of 0.19 indicates a lower risk rather than an increased risk. An odds ratio larger than one indicates an elevated risk.

The results presented do not support the conclusion stated in the abstract. What issues were discovered? The abstract does not even tell readers about the percentage of poor quality ANC recorded in the study.

Introduction

“Among these all deaths, 98% of the stillbirths occur in developing countries”……this is not correct. I guess the pronoun “these” referred to the still births. Therefore, “these all deaths” as used by the authors does not convey the same meaning as stillbirths, which is more specific on the type of death. It is rather 98% of still births that occur in developing countries.

“In Sub-Saharan Africa (SSA) has the highest stillbirth rate globally 28.3 per 1000 births…” the sentence needs a revision as it is grammatically incorrect. What is the information being provided here?

“Therefore, it was important to conduct the effect of quality antenatal care on birth outcome (antepartum still birth)”. How is this possible?

The introduction has to be put into context.

Materials and methods

Why the use of upper cases and lower cases in the heading here?

“From my previous research outcome published in Hindawi Journal,(15) among 1123 mothers participated in the study, about 980 completed the follow up and know their status whether they have live birth or not”. This current paper is written by many authors. Who does the “my” refer to in this case? There is a problem with the sentence structure. What are the authors communicating here? Among 1123 mothers participated in the study is not good grammar

“Hence, had the mothers obtained good quality of antenatal care, they would have delivered in health facilities.”….

I think it's unfortunate that the authors believe this. In general, in SSA, there is a disconnect between the use of ANC and delivery at health facilities; women use ANC but deliver at home. Although the use of ANC enhances the likelihood of delivery in a health facility, this is not always the case. Many factors, not simply ANC, influence the location of delivery. Some elements, such as the commencement of labour and delivery, are out of the individual's control. I'm not convinced the authors aim to draw the conclusion that all women who give birth in health facilities get good ANC and all women who give birth at home get bad ANC. The sentence should be removed or altered, in my opinion.

Independent variable is quality of ANC. The authors state that the women who received ≥ 75% of the essential ANC services… Which are these services that were used to measure quality? How many are they? How was the information collected from respondents? The details are not stated. Do readers have to refer to the publication in Hindawi before they can understand the current study? Obviously not. I think the main exposure variable and how it was constructed has to be made clear in the current study.

Data analysis.

Based on Hosmer and Lemeshow applied logistic regression guide a p-value < 0.2 was considered to select…

Setting criteria for variable exclusion in an adjusted model is intended to ensure model stability, especially when dealing with a large number of exposure variables. Although it has apparent advantages, one drawback of this technique is that significant variables are "thrown away" simply because they do not fulfil the inclusion criteria. One thing to keep in mind when working with datasets is that statistical significance is determined in part by sample size. Because there are few variables in this study, such a criterion may cause complications.

What were the ethical considerations in the current study?

Results

The descriptive information is problematic and with many grammar problems. Authors need to check again. For instance what do they mean by “Pregnancy greater than 34”. I don’t see anything like that in Table 1. Also, some of the percentages (e.g. Age variable) do not sum up to 100. Is the age in years or year?

What do authors mean by “mean pregnancy”?

“Almost half 445 (45.4%) were attempted abortion at least 1 times (Figure 1)”… This is not standard English.

The past obstetric history could be presented in a table too

What is the distribution of quality ANC in the current study? How many women received good or poor quality ANC? Where are the responses from women that were used to compute the level of quality of ANC?

What is the relevance of the “S/N” in the tables? Why does Table 2 start from S/N 11 before 2?

Define the abbreviations used in the tables

If the odds ratio for quality ANC are indeed correct, then the interpretation is wrong. The correct result from the odds ratio of 0.19 means that women who received poor quality ANC were less likely to experience antepartum stillbirth

The variable educational status has small number of observations for some of the subgroups, which has resulted in wider confidence intervals. Is it not possible to collapse some of these groups? How is “read and write” different from grade 1-6 or grade 7-8?

Discussion and conclusion

I expect these two sections to change based on how the main results are interpreted.

Limitation

I'm not sure why the writers' statement is considered a limitation. When women who deliver outside of a health facility are included, the coverage is broader than when only women who birth in a health facility are included.

6. PLOS authors have the option to publish the peer review history of their article (what does this mean?). If published, this will include your full peer review and any attached files.

**Do you want your identity to be public for this peer review?** For information about this choice, including consent withdrawal, please see our Privacy Policy.

Reviewer #1: **Yes: **Michael Boah, PhD

---

## [Decision Letter · Decision Letter 1]

14 Dec 2022

DOES QUALITY OF ANTENATAL CARE INFULENCE ANTEPARTUM STILL BIRTH?IN HOSSANA CITY, SOUTH ETHIOPIA ?

PGPH-D-22-00340R1

Dear Dr Berhe,

We are pleased to inform you that your manuscript 'DOES QUALITY OF ANTENATAL CARE INFULENCE ANTEPARTUM STILL BIRTH?IN HOSSANA CITY, SOUTH ETHIOPIA ?' has been provisionally accepted for publication in PLOS Global Public Health.

Best regards,

Ramachandran Thiruvengadam, M.D.,

Academic Editor

Reviewer Comments (if any, and for reference):

Reviewer's Responses to Questions

**Comments to the Author**

1. If the authors have adequately addressed your comments raised in a previous round of review and you feel that this manuscript is now acceptable for publication, you may indicate that here to bypass the “Comments to the Author” section, enter your conflict of interest statement in the “Confidential to Editor” section, and submit your "Accept" recommendation.

Reviewer #2: All comments have been addressed

2. Does this manuscript meet PLOS Global Public Health’s publication criteria? Is the manuscript technically sound, and do the data support the conclusions? The manuscript must describe methodologically and ethically rigorous research with conclusions that are appropriately drawn based on the data presented.

Reviewer #2: Yes

3. Has the statistical analysis been performed appropriately and rigorously?

Reviewer #2: Yes

4. Have the authors made all data underlying the findings in their manuscript fully available (please refer to the Data Availability Statement at the start of the manuscript PDF file)?

Reviewer #2: Yes

5. Is the manuscript presented in an intelligible fashion and written in standard English?

Reviewer #2: Yes

6. Review Comments to the Author

Reviewer #2: Nil.

7. PLOS authors have the option to publish the peer review history of their article (what does this mean?). If published, this will include your full peer review and any attached files.

**Do you want your identity to be public for this peer review?** For information about this choice, including consent withdrawal, please see our Privacy Policy.

Reviewer #2: No
